

# Using affective knowledge to generate and validate a set of emotion-related, action words

Emma Portch[1], Jelena Havelka[1], Charity Brown[1] and
Roger Giner-Sorolla[2]

[1] School of Psychology, University of Leeds, Leeds, UK
[2] School of Psychology, Keynes College, University of Kent, Canterbury, UK

Corresponding author
Emma Portch,
e.s.portch@leeds.ac.uk

## ABSTRACT

Emotion concepts are built through situated experience. Abstract word meaning is grounded in this affective knowledge, giving words the potential to evoke emotional feelings and reactions (e.g., *Vigliocco et al., 2009*). In the present work we explore whether words differ in the extent to which they evoke 'specific' emotional knowledge. Using a categorical approach, in which an affective 'context' is created, it is possible to assess whether words proportionally activate knowledge relevant to different emotional states (e.g., 'sadness', 'anger', *Stevenson, Mikels & James, 2007a*). We argue that this method may be particularly effective when assessing the emotional meaning of action words (e.g., *Schacht & Sommer, 2009*). In study 1 we use a constrained feature generation task to derive a set of action words that participants associated with six, basic emotional states (see full list in Appendix S1). Generation frequencies were taken to indicate the likelihood that the word would evoke emotional knowledge relevant to the state to which it had been paired. In study 2 a rating task was used to assess the strength of association between the six most frequently generated, or 'typical', action words and corresponding emotion labels. Participants were presented with a series of sentences, in which action words (typical and atypical) and labels were paired e.g., "If you are feeling 'sad' how likely would you be to act in the following way?" ... 'cry.' Findings suggest that typical associations were robust. Participants always gave higher ratings to typical vs. atypical action word and label pairings, even when (a) rating direction was manipulated (the label or verb appeared first in the sentence), and (b) the typical behaviours were to be performed by the rater themselves, or others. Our findings suggest that emotion-related action words vary in the extent to which they evoke knowledge relevant for different emotional states. When measuring affective grounding, it may then be appropriate to use categorical ratings in conjunction with unimodal measures, which assess the 'magnitude' to which words evoke feelings (e.g., *Newcombe et al., 2012*). Towards this aim we provide a set of emotion-related action words, accompanied by generation frequency and rating data, which show how strongly each word evokes knowledge relevant to basic emotional states.

## INTRODUCTION

Emotion words are not just 'words.' Recent theories of semantic representation suggest that abstract words (including emotional words) are predominantly understood due to their grounding in situated experience (e.g., *Vigliocco et al., 2009*; *Wilson-Mendenhall et al., 2011*; *Vinson, Ponari & Vigliocco, 2014*). Words derive meaning because they are bound to the emotional experiences that they refer to; words have the power to re-activate or evoke these internal feelings or states (*Wilson-Mendenhall et al., 2011*). For example, the word 'Justice' is understood because it easily evokes certain emotional connotations, such as feelings commonly associated with receiving a jury verdict (e.g., joy, frustration, dismay; example taken from *Newcombe et al., 2012*). This parallels the proposed situated conceptualisation of concrete words (e.g., 'pen'), which predominantly find grounding in their sensorimotor bindings (e.g., what a 'pen' looks like and how we interact with this object in our environment e.g., *Barsalou, 1999*; *Barsalou et al., 2008*).

Several findings support the proposed distinction in the types of knowledge that ground abstract and concrete words (e.g., *Vigliocco et al., 2009*; *Wiemer-Hastings & Xu, 2005*; *Newcombe et al., 2012*). Using a feature generation task, *Wiemer-Hastings & Xu (2005)* showed that participants generated a significantly higher number of 'experience' and feeling-related properties when defining abstract concepts in comparison to concrete concepts, which elicited a larger proportion of 'entity' and 'situational' properties. Findings from lexical decision tasks also show that dimensional, affective ratings (valence and arousal, e.g., *Lang, 1980*) are better predictors of abstract than concrete word recognition (*Kousta, Vinson & Vigliocco, 2009*; *Kousta et al., 2011*). In contrast to the classic finding, abstract words were processed faster than concrete words when other types of experiential property were controlled (e.g., imageability ratings for each word). Importantly, *Vinson, Ponari & Vigliocco (2014)* also show that valence is similarly predictive of lexical decision latencies when participants responded both to emotion labels (e.g., 'sad') and emotion-related words (e.g., 'death'), showing that affective knowledge provides an important binding for words possessing both strong and weak associations to emotional concepts (see also *Altarriba & Basnight-Brown, 2010*).

These findings suggest that dimensional ratings (valence and arousal e.g., *Lang, 1980*) effectively quantify the affective knowledge that ground abstract words (e.g., *Kousta, Vinson & Vigliocco, 2009*; *Kousta et al., 2011*; *Vinson, Ponari & Vigliocco, 2014*). They build on a large body of previous work, showing that dimensional ratings well predict differences in neutral vs. emotional word processing, using the lexical decision paradigm (e.g., *Estes & Adelman, 2008*; *Larsen et al., 2008*; *Kousta, Vinson & Vigliocco, 2009*), even when very large sets of words are sampled (e.g., *Vinson, Ponari & Vigliocco, 2014*; *Kuperman, 2014*; *Kuperman et al., 2014*). However, some researchers explore the utility of new variables. In particular, *Newcombe et al. (2012)* developed a semantic richness measure, called 'emotional experience.' This measure is similar to body-object interaction (*Siakaluk et al., 2008*) and imageability scales (*Schock, Cortese & Khanna, 2012*), which assess the extent to which words represent and elicit the experiential properties of their referents (in the latter case, sensory and motor properties). As such, the emotional experience variable captures

the ease with which a word evokes affective knowledge. *Newcombe et al. (2012)* collected participant-generated emotional experience ratings for a large set of nouns, predictably finding that emotional experience ratings were higher for abstract than concrete words (*Moffat et al., 2014*; *Vigliocco et al., 2009*).

Importantly, subsequent work showed that emotion experience ratings were significant predictors of abstract word processing in a variety of tasks (e.g., *Siakaluk, Knol & Pexman, 2014*; *Moffat et al., 2014*). For example, participants were significantly slower to process abstract words rated high (vs. low) in emotional experience in a Stroop task, suggesting a larger degree of interference when processing words with strong links to a potentially large pool of affective information. Additionally, when participants engaged in a semantic categorisation task, in which they categorised either 'abstract' or 'concrete' words in a continuous stream, high emotional experience ratings were related to facilitative effects in the abstract categorisation task, and smaller inhibitory effects in the concrete categorisation task (*Newcombe et al., 2012*; *Moffat et al., 2014*). Importantly, emotional experience ratings continued to predict all three effects, even when valence and arousal ratings were entered as predictors (*Newcombe et al., 2012*; *Siakaluk, Knol & Pexman, 2014*; *Moffat et al., 2014*). Taken together, these findings show that emotional experience ratings provide a valid way to quantify affective, experiential knowledge (e.g., *Newcombe et al., 2012*).

Categorical ratings present a related way to assess affective grounding. Arguably, emotional experience ratings provide an 'undifferentiated' quantification, suggestive of the ease with which a word evokes knowledge relevant to a range of emotional states e.g., the word 'funeral' may strongly evoke feelings relevant to different emotions, like 'sadness,' 'anguish' and 'fear' (*Newcombe et al., 2012*). Put another way, emotional experience ratings quantify the magnitude to which a word makes you 'feel' (*Siakaluk, Knol & Pexman, 2014*). However, we might be explicitly interested in the likelihood that the word 'funeral' evokes feelings of 'sadness,' in proportion to feelings and knowledge related to other possible states, like 'fear' and 'anguish.' By posing the emotional label of 'sad(ness),' a constrained 'affective context' is created, under which participants' judge the specific relationship between the emotional concept of 'sadness,' and the word 'funeral.' This conceptualisation brings a greater degree of specificity to the notion that words evoke feelings. In this sense, categorical ratings are similar to context availability measures, which assess the likelihood that words evoke contexts (or, emotional states), in which their referents appear (e.g., *Schwanenflugel & Shoben, 1983*; *Wiemer-Hastings & Xu, 2005*; *Moffat et al., 2014*). Additionally, categorical ratings support *Pecher, Boot & Van Dantzig*'s (*2011*) view of how abstract concepts are grounded. They emphasise that abstract words likely reactivate very specific contexts or situations that we have experienced, rather than being generally evocative; just as concrete words reactivate specific sensory and motor representations, in the same neural areas that initially process sensorimotor information (e.g., *Wilson-Mendenhall et al., 2011*).

Several researchers already provide categorical ratings for emotional words (e.g., *Stevenson, Mikels & James, 2007a*; *Briesemeister, Kuchinke & Jacobs, 2011a*). In *Stevenson, Mikels & James*' (*2007a*) study participants rated each word in the ANEW database (*Bradley &*
*Lang, 1999*), based on extent of association with the basic states of 'happiness,' 'sadness,' 'anger,' 'fear' and 'disgust' (e.g., *Ekman, 1992*). Here discrete emotional states, denoted by a label, create a constrained 'affective context' and participant ratings indicate the likelihood to which each ANEW word proportionally evokes knowledge relevant to those emotional states. *Stevenson, Mikels & James (2007a)* were particularly interested in whether words could be 'discretely' related to a particular emotion label. Given that words likely evoke experiential knowledge relevant to a number of emotional states (e.g., *Siakaluk, Knol & Pexman, 2014*), we reframe *Stevenson, Mikels & James (2007a)* terminology to talk about 'disproportional' relationships (e.g., how strongly does a word evoke knowledge relevant to one basic emotion state, in comparison to others?).[1] *Stevenson, Mikels & James (2007b)* assumed that a disproportional association was present when the rating given for the word/label pair was one standard deviation higher than ratings given to that word when paired with all other emotion labels. Using this method 44.54% of the 1,034 words tested were disproportionally related to one or two discrete emotion labels. *Briesemeister, Kuchinke & Jacobs (2011a)* produced similar findings when using this rating method with German nouns included in the Berlin Affective Word List (*Võ, Jacobs & Conrad, 2006*; *Võ et al., 2009*). When *Stevenson, Mikels & James (2007a)* criterion was applied, 25.18% of the words within DENN-BAWL could be disproportionately associated with a particular emotion label.

Importantly, subsequent work shows that categorical ratings for both English and German words predicted lexical decision latencies (e.g., *Briesemeister, Kuchinke & Jacobs, 2011a*; *Briesemeister, Kuchinke & Jacobs, 2011b*; *Briesemeister, Kuchinke & Jacobs, 2014*). In particular, words disproportionately related to the discrete state of 'happiness' were processed faster than neutral words and words disproportionately associated with negative discrete categories, like 'disgust,' 'fear' (*Briesemeister, Kuchinke & Jacobs, 2011a*) and 'anger' (*Briesemeister, Kuchinke & Jacobs, 2011b*). *Briesemeister, Kuchinke & Jacobs (2014)* and *Briesemeister et al. (2014)* both provide evidence to suggest that behavioural facilitation was not simply driven by the positive valence of these words. Temporally dissociable ERP components (*Briesemeister, Kuchinke & Jacobs, 2014*) and topographically distinct brain activity (*Briesemeister et al., 2014*) were found when participants processed words that differed in 'happiness' association (high vs. low), but were matched on valence and arousal.

A recent study by *Westbury et al. (2014)* provides further support for the categorical approach. Rather than using participant ratings, *Westbury et al. (2014)* mapped the semantic distance between emotion labels and words, based on how frequently they co-occurred in close proximity within a large corpus of text (HiDeX; e.g., *Shaoul & Westbury, 2010*). According to *Vigliocco et al.*'s (*2009*) theory of semantic representation, linguistic co-occurrence supplements experiential grounding of abstract words, pairing affective components whose referents we may not have directly experienced (e.g., knowing that funerals evoke feelings of sadness arguably relies on having attended a funeral; see also *Barsalou et al., 2008*). First, *Westbury et al. (2014)* found that the dimensional ratings for a large subset of words (*Warriner, Kuperman & Brysbaert, 2013*) could be partially predicted by the quantified linguistic co-occurrence between those words and an accepted

[1] We acknowledge that the word 'categorical' has strong, dichotomous connotations; something is either part of a category, or it is not. Although we argue for a proportional, rather than a categorical, interpretation we continue to use the word 'categorical' to describe our approach due to its strong relationship with other work that has used this terminology e.g., *Stevenson, Mikels & James (2007a)*; *Briesemeister, Kuchinke & Jacobs (2011a)*.

set of emotion labels. Second, they found that these co-occurrence values could be used to predict lexical decision latencies for those words (taken from the English Lexicon Project, *Balota et al., 2007*). In some cases, co-occurrence values were better predictors of latency than valence and arousal ratings, particularly when considering co-occurrence with the emotion labels 'pleasant' and 'unpleasant.'

These investigations suggest that categorical ratings, or measures which quantify the proportional association between emotion labels and words, are useful for characterising the way abstract words are processed. Although some findings may be interpreted in a way to suggest that categorical ratings capture different aspects of emotional word processing than standard dimensional variables (e.g., *Stevenson, Mikels & James, 2007a*; *Briesemeister, Kuchinke & Jacobs, 2011b*; *Briesemeister, Kuchinke & Jacobs, 2014*; *Briesemeister et al., 2014*; *Westbury et al., 2014*) it is beyond the scope of the present work to assess the relationship between, or relative merits of the two approaches (see also *Newcombe et al., 2012*). Importantly, though, we do suggest that categorical ratings may be particularly useful for quantifying the affective grounding of emotion verbs, or action-related words. Here we single out words which describe behaviours related to particular emotional states, without naming the emotion itself (*Pavlenko, 2008*). It is not yet possible to test this proposal as studies using the DENN-BAWL focus exclusively on emotional nouns (*Briesemeister, Kuchinke & Jacobs, 2011a*; *Briesemeister, Kuchinke & Jacobs, 2014*). Further, it is unclear whether nouns, adjectives and verbs were equally sampled when *Briesemeister, Kuchinke & Jacobs (2011b)* selected words from *Stevenson, Mikels & James*' (*2007a*) categorisation of the ANEW, or when *Westbury et al. (2014)* sampled from HiDeX (e.g., *Shaoul & Westbury, 2010*).

We argue that emotion-related action words hold a special kind of relationship with experiential knowledge. On the one hand these words may be classified as 'concrete.' According to *Vigliocco et al.*'s (*2009*) framework then, verb meaning should be predominantly situated in sensorimotor knowledge and understood by reactivation in visual and motor areas (e.g., *Pulvermüller, 1999*; *Hauk, Johnsrude & Pulvermüller, 2004*). In support, various researchers show that processing of words directly related to emotional expressions and behaviours e.g., 'smile,' activate face and body-specific regions for performing that action (e.g., *Niedenthal et al., 2009*; *Moseley et al., 2012*) and improve understanding of these expressions, when shown by actors (e.g., *Foroni & Semin, 2009*; *Halberstadt et al., 2009*). On the other hand, verbs that refer to emotional actions are still 'emotional' in nature (*Altarriba & Basnight-Brown, 2010*; *Vinson, Ponari & Vigliocco, 2014*). *Wilson-Mendenhall et al. (2011)* emphasise that affective, experiential knowledge is necessarily multi-faceted, as it is built within the context of situated activity, and thus includes various actions and bodily sensations. Therefore, words referring to emotional actions are likely grounded in both sensorimotor and affective, experiential knowledge.

Due to their dual-experiential-representation, it may be important to make an 'affective context' salient when attempting to measure the affective grounding of words that refer to emotional actions. This additional step is less necessary when presenting more abstract emotional words, such as nouns, which have weaker sensorimotor grounding (e.g., *Vigliocco et al., 2009*). Nouns like 'cancer,' 'death' and 'funeral' are likely to

spontaneously evoke unambiguous, negative affective knowledge, even when presented in isolation (e.g., *Pavlenko, 2008*; *Vinson, Ponari & Vigliocco, 2014*), which makes it highly appropriate to use standard dimensional or emotional experience ratings to capture their emotional meaning (e.g., *Newcombe et al., 2012*). However, when the verb 'jump' is presented alone several alternative, but equally acceptable emotional interpretations are available, as the word has both positive and negative connotations. For example, while someone might 'jump for joy,' they may also jump in reaction to a surprising or fearful stimulus.

Physiological evidence supports the notion that it is comparatively difficult to extract emotional meaning from isolated verbs. Comparing across paradigms, the event-related potentials commonly associated with early and late semantic processing of single emotional words (e.g., *Herbert et al., 2006*) are commonly evidenced at a later onset for emotional verbs (*Schacht & Sommer, 2009*; *Palazova et al., 2011*) than for emotional nouns (e.g., *Kanske & Kotz, 2007*; *Kissler et al., 2007*) or adjectives (*Herbert et al., 2006*; *Herbert, Junghöfer & Kissler, 2008*).

With reference to the previous example, emotional meaning is easier to interpret when more information is available to provide an 'affective context' e.g., if we know that the actor jumped because 'the car crashed into the nearby lamppost.' In this case, the 'jump(ing)' behaviour is likely related to a negative emotional state, most likely to be 'fear.' In support, *Schacht & Sommer (2009)* reported Early Posterior Negative (EPN) and Late Positive Complex (LPC) onsets comparable to those for emotional nouns and adjectives when a clear, 'affective context' was applied. Here participants responded to a verb preceded by a noun (e.g., 'lover-kiss'). *Schacht & Sommer (2009)* argue that the preceding noun improved participants' ability to extract the intended, emotional meaning from test verbs during a lexical decision task. Applying a similar manipulation, *Palazova, Sommer & Schacht (2013)* found comparable EPN onsets when emotional verbs referred to more concrete, context-invariant behaviours, which had clear affective connotations (e.g., to dance vs. to hope).

The present work aims to explore whether a categorical approach can be used to examine the affective, experiential knowledge that partially grounds action word meaning. Importantly, in the first study we pose basic emotion labels (e.g., 'sad') to create a constrained 'affective context.' Participants will self-generate emotional action words that they commonly associate with each emotional state. Generation frequencies, per action word, will be indicative of the likelihood that the word evokes affective, experiential knowledge relevant to paired emotion labels. In the second study a rating task will be conducted to validate use of generation frequencies as a measure of associative strength. Action words are paired with the emotional labels to which they have been most disproportionately generated, and rated according to the strength of that association.

This work provides relevant research communities (e.g., researchers interested in both emotion and language processing) with a database of emotion action words. Accompanying generation frequency (study 1) and rating data (study 2) are suggestive of the extent to which these words evoke affective knowledge related to a set of basic emotional states.

# STUDY 1- IDENTIFYING ACTION WORDS THAT PROPORTIONALLY EVOKE AFFECTIVE KNOWLEDGE

In study 1 we use emotion labels to provide a constrained, 'affective context.' Following *Stevenson, Mikels & James (2007a)* and *Briesemeister, Kuchinke & Jacobs (2011a)*, we present the universal, basic emotion labels used by *Ekman (1992)*; 'happy,' 'sad,' 'fear,' 'anger,' 'disgust' and 'surprise.' We reason that these states represent commonly experienced emotions which will be fluently associated with behavioural referents.

Rather than use a rating task, we conduct a highly constrained semantic feature-generation task. Participants are instructed to self-generate multiple single-word actions that they commonly associate with experiencing each of these discrete emotional states (see *McRae et al., 2005*; *Vinson & Vigliocco, 2008*; *Buchanan et al., 2013* for broader examples of semantic feature generation[2]). Explicit instructions were important as action words have rarely been produced when emotion labels are posed as concepts in feature generation tasks (e.g.; *Hutchison et al., 2010*; *Buchanan et al., 2013*). By encouraging participants to engage separately with each emotion label we also hope to widen the stimulus set, as rating methods often produce a 'happiness asymmetry' (many words are strongly associated with 'happiness,' but far fewer words are associated with discrete, negative states e.g., *Stevenson, Mikels & James, 2007a*; *Briesemeister, Kuchinke & Jacobs, 2011a*).

Overall, we measure the likelihood that an action word evokes discrete affective knowledge based on the frequency of participants who endorse the pair (e.g., *McRae et al., 2005*). However, we acknowledge that the ability to infer proportional association also relies on the number of additional emotional states to which the action word is generated.

## METHOD

### Ethics

This research is subject to ethical guidelines set out by the British Psychological Society (1993) and was approved by the School of Psychology's ethics committee, at the University of Leeds (reference number: 13-0032, date of approval: 24/02/2013).

### Participants

Twenty-five participants (17 female, 8 male) generated action words. Participants had a mean age of 27.24 (SD=7.63) and all reported themselves to be native English speakers (7 participants spoke a second language, though did not consider themselves fluent). An opportunity recruitment method was used. Participants responded to links posted on research recruitment websites and completed the study online (e.g., http://www.psych. hanover.edu/research/exponnet.html; http://www.onlinepsychresearch.co.uk; http:// www.in-mind.org/content/online-research; http://www.reddit.com/r/SampleSize).

### Procedure

All materials, including informed consent items, were presented using the Survey Monkey platform (http://www.surveymonkey.com, Survey Monkey Inc. Palo Alto, California, USA). Participants ticked a series of boxes to confirm that they understood

[2] We acknowledge that similar methods have been used to elicit related stimuli, such as action-readiness and tendency items (*Smith & Ellsworth, 1985*; *Frijda, 1986*; *Frijda, Kuipers & Ter Schure, 1989*). However, these items usually refer to a general anticipatory state that the individual enters after appraising an emotionally salient event (*Frijda, 1986*). Although important components of affective knowledge, these items are generally dissociable from the concrete, overt behaviours derived in the present study, which may be viewed as the eventual behavioural consequence of experiencing such states.

task instructions and gave their informed consent to take part. Participants were then asked to carefully read the definition of an emotion-related action word, below (taken from *Pavlenko, 2008*). Definitions were edited to include relevant examples.

*'Emotion-related' words are used to describe behaviours related to a particular emotional state, without naming the actual emotion. For example, the word 'cry' might describe the behaviour of someone feeling sad while the word 'smile' may describe the behaviour of somebody who is happy.'*

Participants were directed to six basic emotion labels, listed below the definition ('sad,' 'happy,' 'anger,' 'disgust,' 'surprise' and 'fear,' *Ekman, 1992*). They were asked to generate as many emotional action words as they could which were related to each basic label. Separate boxes were provided for participants to type their examples. Participants were instructed to provide single-word answers and to avoid label synonyms or adverbs (e.g., 'sadness,' 'sadly'). They were also discouraged from using the internet to generate responses. Participants were asked to work on the basic labels sequentially and labels were presented in a randomised order across participants. There was no time limit imposed on word generation.

## RESULTS: DATA MODIFICATIONS AND MODAL EXEMPLARS

In total, participants generated 362 unique words, across the six labels. On average, participants each generated 27.32 words during the task (SD = 15.18). We parsed the data in various ways to determine an acceptable set of action words, which were 'modally' associated with one or more emotion labels (see *McEvoy & Nelson, 1982*; *Doost et al., 1999*; *McRae et al., 2005*; *Vinson & Vigliocco, 2008* for similar methods). The Cambridge Online English Dictionary (http://www.dictionary.cambridge.org/) and an online Thesaurus (http://www.Thesaurus.com) were consulted to support these modifications. First, words were deemed unacceptable if (a) they did not describe a concrete action (e.g., 'tearful'; both verbs and nouns were accepted), or (b) were synonyms for the emotion label itself (e.g., 'afraid,' generated in response to 'fear'). Second, multiple-word responses or phrases were only retained if they could be simplified to a single word with the same or similar meaning, for example, 'sharp intake or breath' was replaced with 'gasp.' Third, merging techniques were used either when participants provided grammatical derivatives or plurals of the same word (e.g., 'ran,' 'run,' 'runs,' 'running,' 'ran away') or generated synonyms for action words that had already been provided by themselves or others (e.g., 'scream' and 'shriek'). In the former case, plurals were changed to their singular form and grammatical derivatives were merged and represented by the simplest version, provided their meaning did not change (e.g., 'run').

The second type of merging (non-derivative words) was wholly motivated by our need to develop stimuli for study 2. Here we required only six action words, each of which held the most disproportional association with one of the six emotion labels, respectively. Therefore, it was important to ensure that words with the same/very similar meanings were grouped together, and their frequencies summed, to aid assessment of how strongly those

3 Although this type of merging helped to identify the top-six modal action words, for use in study 2, it necessarily inflated the apparent frequency-based strength of association between those core action words and corresponding emotion labels. Readers are encouraged to consult Appendix S1, in which all modal exemplars are listed alongside unmerged generation frequencies, which provide a clearer estimation of the strength with which individual action works evoke affective knowledge relevant to different emotion states. From Appendix S1, researchers may select stimuli based on unmerged exemplars, or apply their own criteria to identify and merge synonymous exemplars.

4 It was particularly difficult to make merging decisions about the exemplar 'cry.' As this exemplar was given in response to the 'sad,' 'anger,' 'fear,' 'happy' and 'surprise' categories, consideration of cue word could result in two (or more) definitions being accepted. To illustrate, when generated in response to 'sad(ness)' the definition 'to weep or make sad sounds' would be most relevant, but when generated in response to 'anger' the definition 'to call out/yell' was most appropriate (definitions taken from http://www. Thesaurus.com). Arguably participants may have had either meaning in mind when they generated the exemplar in response to the remaining emotion labels, which complicated the issue. We made the decision to merge 'cry' contingent on the first sadness-related definition, only, as the exemplar was most frequently given in response to the 'Sad' category. 'Cry' become the core action word, and 'weep' and 'sob' the subsidiary action words. As 'cry' was already the unmerged, top modal exemplar for 'sad(ness),' this merging decision did not change the modal response that was chosen for the 'sad' label in study 2. If we had alternatively (or additionally) chosen to merge according to the second definition, 'cry' could have been grouped with 'scream,' 'shout' and 'shriek.' This was problematic as our criteria suggested that 'scream' and 'shriek' could be merged with 'yell,' but 'yell' could not be merged with 'cry.' Therefore, the strategy adopted was both simpler, and more conservative.

related behaviours evoked discrete, affective knowledge.[3] Strict criteria were imposed for this form of merging. Action words were only classed as synonymous if there was evidence of forward and backward association e.g., when 'laugh' was entered into the thesaurus 'giggle' was given as a synonym, and when 'giggle' was entered into the thesaurus, 'laugh' was given as a synonym. We were mindful that some action words could have multiple meanings when presented in isolation (e.g., *Schacht & Sommer, 2009*). For example, the action word 'jump' could mean 'to leap, spring or skip,' 'to recoil' or 'to avoid' (definitions taken from http://www.thesaurus.com). In these cases the participants' intended meaning was discerned by considering the emotion label to which the word had most frequently been generated. As the word 'jump' was frequently endorsed for the labels 'surprise' and 'fear' it went unmerged with 'skip,' which although a synonym, was only given in response to the label 'happy.' Here we considered that the two words likely had a different intended meaning, each congruent with the core emotion concept to which they had been modally generated (see *Buchanan et al., 2013* for similar consideration of 'cue' word when merging 'target' words).

Where merging occurred, frequencies for both/all action words were added together. For non-derivative synonyms the dominant response was retained, based on existing frequencies (i.e., the action word given by the highest number of participants). This exemplar became the 'core' action word and non-dominant responses were subsumed and became 'subsidiary' action words. For example, in response to the label 'sad,' 'cry' became a core action word and the synonyms 'weep' and 'sob' became subsidiaries.[4] The number of participants who generated the action words 'cry,' 'weep' and 'sob' were added together to provide a frequency total for the core action word ('cry'). Note that frequencies could exceed 25 if participants had provided both core and subsidiary action words in response to the same emotion label.

Following these steps our set still contained a large number of 'idiosyncratic' responses, generated by only one participant in response to a particular label (124 words, 56.88% of remaining responses). These exemplars are unlikely to represent words which commonly evoke discrete affective knowledge; therefore, we decided to remove these responses from the sample (see *Buchanan et al., 2013*). Following removal of idiosyncratic responses, there were 51 unique, modal action words, including 15 core action words, and 19 subsidiary action words. Here 'modal' refers to an action word that was generated by two or more participants, but was not synonymous with other responses and went unmerged. Therefore, they differ from 'core' and 'subsidiary' action words. This final selection represents 14% of the total number of unique words originally generated.

The top three most frequently generated action words, per emotion label, are shown in Table 1. Response frequencies are shown in parenthesis, in the second column. When these words represent core exemplars, frequencies also include the number of participants who generated subsidiary action words (corresponding subsidiary words are shown in the column three). Frequencies above 25 are shown when a proportion of participants gave both the core and subsidiary exemplars in response to the same emotion label. The full set of action words (core, subsidiary and modal), are provided in Appendix S1. In addition,

**Table 1  Top three, most frequently generated action words for each emotion label.** Action words are presented alongside subsidiary responses (where appropriate). Response frequencies for each action word are presented within parenthesis in the second column. These frequencies represent merged totals when a corresponding subsidiary action word is shown in the third column.

| Emotion label | Most frequent action words (response frequency) | Corresponding, subsidiary action words (core action word) |
|---|---|---|
| Anger | Scream (34); Hit (13); Cry (7) | Shout/Yell/Shriek (Scream); punch (Hit); sob/weep (Cry) |
| Happy | Smile (27); Laugh (20); Dance (10) | Grin (Smile); Giggle (Laugh); Skip (Dance) |
| Sad | Cry (23); Frown (9), Withdraw (7) | Sob/Weep (Cry); Grimace (Frown) |
| Disgust | Recoil (7); Frown (6); Gag/Vomit (5 each) | Cringe (Recoil); Grimace (Frown); Retch (Gag) |
| Fear | Hide/Run (13 each); Shiver (11); Cry (9) | Avoid (Hide); Shake (Shiver); Sob/Weep (Cry) |
| Surprise | Jump (15); Gasp (13); Scream (12) | Inhale/Sharp Intake (Gasp); Shout/Yell/Shriek (Scream) |

all responses are provided in the Supplemental Information (acceptable and unacceptable idiosyncratic and modal responses).

Analysing by exemplar, 78.43% of all modal action words were generated in response to one emotion label only, leaving 21.57% that were generated for multiple labels. This distinction was present even for the most frequently generated action words, displayed in Table 1. When only these exemplars were considered, 15.79% represented the most frequent responses for more than one emotion label, and 68.75% were generated by at least two participants in response to one of more other emotion labels. These findings support the work of *Stevenson, Mikels & James (2007b)*. In their study, although 44.54% of ANEW words obtained ratings to suggest that they were disproportionately associated with one (or two) discrete emotions, 22.70% of words were associated with three or more emotion labels, representing an analogue to the 'overlapping' exemplars in the present study.

## DISCUSSION

In the present study we introduced a constrained 'affective context' to identify action words that were likely to evoke affective knowledge, proportionally relevant to different emotional states (e.g., *Stevenson, Mikels & James, 2007a*). The greater the number of participants that generated a particular action word in response to an emotion label, the greater likelihood that that action word would be situated in, and evoke affective knowledge relevant to that emotion. Both action words and generation frequencies are available in Appendix S1. We suggest possible uses for our stimuli in the general discussion.

Importantly, findings suggest that participants generated a selection of action words that were either strongly (or disproportionately) associated with a particular emotional state, or were proportionally related to a number of different emotional states (overlapping exemplars). These findings have important implications both for theories of affective, experiential grounding and emotional attribution; the latter addressed in the general discussion. Some researchers suggest that words are understood by evoking very specific representations of situations in which their referents appear (e.g., *Schwanenflugel & Shoben, 1983*; *Pecher, Boot & Van Dantzig, 2011*). This parallels understanding of concrete concepts, which rely on reactivation in the same sensorimotor areas initially recruited during interactions with the referent object (e.g., *Barsalou, 1999*). Finding that some

action words were disproportionately associated with one emotion label appear to provide support for this view. However, finding overlapping exemplars support the notion that words are generally evocative and have the potential to re-activate affective knowledge relevant to a range of emotional states (e.g., *Newcombe et al., 2012*). In the present study 'cry' may be a particularly good example of a word that is 'generally' evocative. This exemplar and its synonyms ('sob' and 'weep') were frequently given in response to the 'sad,' 'anger' and 'fear' labels, and also by a smaller number of participants in response to the 'happy' and 'surprise' labels. In study 2 we use a rating task to assess the robustness of the most frequent action word-to-label associations, generated during study 1.

## STUDY 2- VALIDATING ASSOCIATIONS BETWEEN ACTION WORDS AND EMOTION LABELS

In study 2 we assess (a) the typicality of self-generated action words, and (b) the stability of action word-to-label associations. We adopt a rating task, similar to *Stevenson, Mikels & James (2007a)*, in which participants rate the relationship between the six most frequently generated action words, and each discrete, emotion label. Emotion labels and action words are presented within a sentence e.g., "if you see someone 'recoil' how likely are you to think that they are feeling the following emotion?... 'disgust.'" Primarily, we would expect ratings to indicate a comparatively stronger association between action words and the emotion labels to which they were (most frequently) generated. This would confirm that the word is understood due to its (dis)proportional activation of affective knowledge relevant to that emotional state.

This validation attempt was particularly important for assessing whether the top exemplars 'cry' and 'smile' were as strongly linked to the respective emotional states of 'sad(ness)' and 'happ(iness)' as generation frequencies suggested. This was a concern as both action word/label pairs had been included as examples in the task instructions for study 1, so frequent endorsement may not reflect spontaneous generation. This may also explain why the word 'cry' was given so frequently, across the different 'affective contexts.' In addition, although participants were discouraged from using the internet to generate their responses during study 1, we were unable to definitively rule out the possibility that they had done so. Use of external sources may have inflated frequencies, artificially creating modal exemplars. Although this seems unlikely, as participants generated a larger number of idiosyncratic than modal exemplars, it is important to address this possible methodological issue.

Two further manipulations were applied to the rating task to test the robustness of action word-to-label associations. First, we varied rating direction (i.e., whether participants made an action word-to-emotion category, or emotion category-to-action word association). The following is an example of an action word-to-category rating: "if you see someone 'cry,' how likely are you to think that they feel 'sad?'" Researchers commonly evaluate semantic relationships by measuring both the 'forward' and 'backward' associations between category labels and exemplars, and quantify the strength of the association using conditional probabilities (e.g., *Nelson, McEvoy & Schreiber, 2004*).

Here conditional probabilities measure whether action words evoke knowledge relevant to a particular emotional state as strongly as that emotional state (label) evokes knowledge of the action word's referent.

Second, we asked participants to rate action word/category pairings from both a first person perspective (e.g., "If you are 'crying,' how likely is it that you are feeling 'sad?'") and a third person perspective. (e.g., "if someone is 'crying,' how likely are they to be feeling 'sad?'"). This was an exploratory manipulation, which had the potential to inform us about the way in which affective knowledge is used for emotional attribution. On the one hand, higher ratings between action words and emotion labels might be expected when a first-person perspective is applied. Given that affective knowledge is predominantly grounded in an individual's situated experience (e.g., *Vigliocco et al., 2009*), words may preferentially evoke feelings that are self-relevant. Conversely, participants may view a simpler correspondence between behaviours and emotions for other people, than for themselves. Self-relevant affective knowledge may be richer and more variable, complicating behaviour-to-state mappings when participants use first-person instructions (e.g., 'people tend to act this way when they are feeling a certain emotion, but when I was feeling happy I didn't act that way'). This account would predict stronger action word/label ratings when participants adopt a third-person perspective.

## METHOD

### Ethics

This research is subject to ethical guidelines set out by the British Psychological Society (1993) and was approved by the School of Psychology's ethics committee, at the University of Leeds (reference number: 13-0032, date of approval: 24/02/2013). As before, informed consent items were embedded in an online survey and participants agreed to take part by ticking a series of boxes.

### Design

A 2 (instruction perspective: first or third person, between) × 2 (rating direction: category to action word or action word to category, between) × 2 (typicality: typical or atypical label/action word pairing, within) mixed factorial design was employed. The instruction perspective factor manipulated whether participants received first-person perspective instructions ("if you are feeling 'sad,' how likely are you to act in the following way?" e.g., 'cry') or third person perspective instructions ("if someone is feeling 'sad,' how likely are they to act in the following way?" e.g., 'cry'). The rating direction factor manipulated whether participants rated associations in an action word-to-category direction ("if you are 'crying,' how likely are you to be feeling the following emotion?" e.g., 'sad') or a category-to-action word direction ("if you are feeling 'sad,' how likely are you to act in the following way" e.g., 'cry'). Participants each made 36 ratings, based on all combinations of six discrete emotion labels and the action words most frequently endorsed in response to each of these labels, during study 1. Feature generation data determined whether emotion

label/action word pairings were typical (e.g., six pairs, 'happy' and 'smile'), or atypical (30 pairs, e.g., 'sad' and 'smile').

Participants were presented with an open-ended sentence for each rating, which included either an emotion label or action-word e.g., "if you are feeling 'sad,' how likely are you to act in the following way?" Participants were invited to substitute each of the six action words (or labels) into the end of this sentence (e.g., 'cry'), and to provide a likelihood rating for each label/action word pairing. After all six ratings were submitted, participants were presented with the next open-ended sentence, which included a new label (or action word). Overall, participants made ratings in six, separate blocks, which presented a different label (or action word) to be rated against each action word (or label), respectively. Block order was counterbalanced across participants. Within a particular block, participants encountered each of the six ratings in a fixed order. Although fixed per participant, this order was randomised per block, to ensure that the typical pairing was not always presented in the same rating position (e.g., in the 'sad' block participants rated associations with action words in the following order: 'smile,' 'cry' 'jump'…, but in the 'happy' block they rated action words in a different order: 'hide,' 'scream,' 'smile'). Therefore, while block order differed, rating order within blocks was the same for all participants within a particular condition.

## Participants

Forty participants each completed the task using first-person perspective instructions (25 female, Mean age = 26.48, SD = 8.97) and third-person perspective instructions (29 female, Mean age = 27.53, SD = 9.47). Forty participants completed tasks that required category-to-action word ratings (31 female, Mean age = 25.65, SD = 9.56) and forty completed tasks that required action word-to-category ratings (29 female, Mean age = 28.35, SD = 8.70).

Participants indicated whether they spoke any languages in addition to English and estimated how many years they had been able to do so. Those judged to be fluent bilinguals or multi-linguals were omitted from the sample. An opportunity recruitment method was used; participants responded online, to links posted on social media sites (see Study 1). The study was presented using the Survey Monkey platform (http://www.surveymonkey.com, Survey Monkey Inc., Palo Alto, California, USA). There was no time limit imposed.

## Materials

We re-used the six basic emotion labels from study 1 ('fear,' 'happy,' 'sad,' 'disgust,' 'anger' and 'surprise,' e.g., *Ekman, 1992*). The most frequently generated action words for each emotion label were selected from the merged, feature generation data. They were as follows: 'scream' (matched with 'anger'); 'smile' ('happy'), 'cry' ('sad'), 'recoil' ('disgust'), 'hide' ('fear') and 'jump' ('surprise').

## Procedure

Each participant was randomly assigned to one of the four between-participants conditions of the 2 (instruction perspective) × 2 (rating direction) design. Ratings were

**Table 2 Table of effects for the instruction perspective × rating direction × emotion category × typicality, mixed factorial ANOVA.** $F$, $p$ and $\eta_p^2$ statistics are given for each effect. Italics denote significant ($p < 0.05$) and marginal ($p < 0.1$) effects.

| Effect | DF | MSE | F | p | $\eta_p^2$ |
|---|---|---|---|---|---|
| *Category* | *(4.34, 325.24)* | *0.53* | *18.93* | *<0.001[*]* | *0.20* |
| *Typicality* | *(1,75)* | *1.04* | *696.35* | *<0.001[*]* | *0.90* |
| *Instruction Perspective* | *(1,75)* | *2.05* | *6.19* | *0.015[*]* | *0.08* |
| *Rating Direction* | *(1,75)* | *2.05* | *5.50* | *0.022[*]* | *0.07* |
| Category × Instruction Perspective | (4.34, 325.54) | 0.53 | 1.42 | 0.23 | 0.02 |
| Category × Rating Direction | (4.34, 325.54) | 0.53 | 0.28 | 0.90 | 0.004 |
| Typicality × Instruction Perspective | (1,75) | 1.04 | 0.08 | 0.77 | 0.001 |
| Typicality × Rating Direction | (1,75) | 1.04 | 1.25 | 0.27 | 0.016 |
| *Category × Typicality* | *(4.25, 318.60)* | *0.45* | *26.79* | *<0.001[*]* | *0.34* |
| Instruction perspective × Rating Direction | (1,75) | 2.05 | 0.37 | 0.55 | 0.005 |
| Category × Instruction Perspective × Rating Direction | (4.34, 325.24) | 0.45 | 0.90 | 0.47 | 0.012 |
| Typicality × Instruction Perspective × Rating Direction | (1, 75) | 1.04 | 1.37 | 0.25 | 0.018 |
| *Category × Typicality × Instruction Perspective* | *(4.25, 318.60)* | *0.45* | *0.90* | *0.47* | *0.0012* |
| *Category × Typicality × Rating Direction* | *(4.25, 318.60)* | *0.45* | *3.84* | *0.004[*]* | *0.049* |
| *Category × Typicality × Instruction Perspective × Rating Direction* | *(4.25, 318.60)* | *0.45* | *3.43* | *0.008[*]* | *0.044* |

**Notes.**

[*] Greenhouse-Geisser corrections were applied for 'Category' and 'Category × Typicality' effects.

made on a five-point Likert-style scale for each question, anchored 'Very Unlikely' (1) to 'Very Likely' (5). All participants were presented with the same combination of emotion label/action word pairings and made 36 ratings in total.

## RESULTS

### Data preparation

For each emotion label, two mean ratings were calculated per participant. The 'typical' mean was the rating given to the most typical label and emotion word pairing, according to the feature generation data (e.g., 'cry' and 'sad'). The five remaining ratings given by the participant were summed and then averaged to produce a grouped 'atypical' score (mean scores for the full set of 36 label/action word ratings are shown in Appendix S2).

### Analysis

A 2 (instruction perspective: first or third) × 2 (rating direction: category-to-action word or action word-to-category) × 6 (category: sad, anger, happy, disgust, surprise, fear) × 2 (typicality: typical or atypical) mixed factorial ANOVA was performed. Instruction perspective and rating direction were between-subjects factors. Main effects and interactions are displayed in Table 2. Hereafter, we focus on interactions with the typicality factor. 'Typicality' reflects the strength of association between action words and emotion labels (operationalised here as high or low), thus indicating the likelihood that action words disproportionately evoke affective knowledge relevant to emotional states.

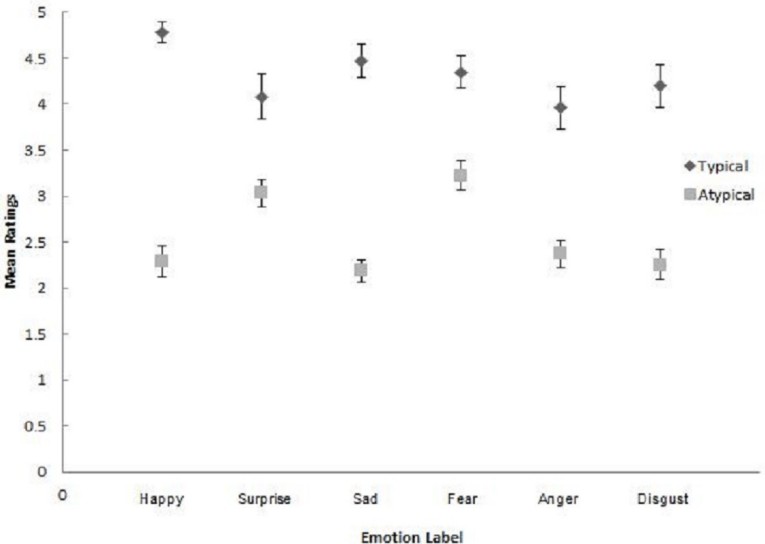

**Figure 1 Mean typical and atypical ratings, per emotion label.** Error bars represent 95% CI.

**Table 3 Mean ratings for typical and atypical word pairs, per emotion category.** $t$, $p$ and $d$ statistics are presented for each emotion category (standard deviations shown in parenthesis).

| Emotion category | Typical mean (SD) | Atypical mean (SD) | $t$ | $p$ | $d$ |
| --- | --- | --- | --- | --- | --- |
| Happy | 4.78 (0.47) | 2.29 (0.75) | 24.24 | <0.001 | 4.0 |
| Surprise | 4.08 (1.11) | 3.03 (0.66) | 9.06 | <0.001 | 1.58 |
| Sad | 4.47 (0.81) | 2.19 (0.55) | 23.86 | <0.001 | 3.31 |
| Fear | 4.35 (0.80) | 3.22 (0.70) | 13.04 | <0.001 | 1.51 |
| Anger | 3.96 (1.02) | 2.37 (0.67) | 13.22 | <0.001 | 1.85 |
| Disgust | 4.20 (1.03) | 2.25 (0.72) | 13.12 | <0.001 | 2.21 |
| *Totals* | *4.31 (0.87)* | *2.56 (0.68)* | - | - | - |

**Notes.**

[*] Degrees of Freedom were always (1,78).

## Validation of self-generation data

Participants gave significantly higher mean likelihood ratings to typical pairings ($M = 4.31$, SD $= 0.56$), than grouped atypical pairings, ($M = 2.56$, SD $= 0.49$), on a scale from 1–5. This finding provides support for the label-action word associations derived from the feature generation data (study 1). The typicality effect was qualified by a significant interaction with emotion category, prompting investigation of the effect for each discrete, emotion (see Table 3 and Fig. 1).

All six paired samples $t$-tests were significant and in the anticipated direction (typical category/action word pairings received higher association ratings than the grouped atypical pairings). Therefore, the interaction likely reflects general differences in the strength with which typical action words evoke affective knowledge disproportionally related to corresponding emotional states, all effects being conventionally large ($d > 0.8$). In support, Fig. 1 shows that the 95% confidence intervals for mean typical ratings and

**Table 4 Mean action word-to-category (A-to-C) and category-to-action word ratings (C-to-A), by typicality and emotion category.** $F, p$ and $\eta_p^2$ statistics are displayed for each effect. Significant interactions are starred ($p < 0.05$).

| Category | Mean C-to-A rating (SD) | | Mean A-to-C rating (SD) | | $F$ | $MSE$ | $p$ | $\eta_p^2$ |
|---|---|---|---|---|---|---|---|---|
| | Typical | Atypical | Typical | Atypical | | | | |
| Happy | 4.77 (0.54) | 2.10 (0.77) | 4.80 (0.41) | 2.47 (0.70) | 2.73 | 0.41 | 0.10 | 0.034 |
| Surprise | 3.97 (1.18) | 2.97 (0.79) | 4.18 (1.03) | 3.08 (0.51) | 0.18 | 0.54 | 0.67 | 0.002 |
| Sad | 4.21 (1.00) | 2.18 (0.59) | 4.73 (0.45) | 2.20 (0.51) | 7.40 | 0.33 | 0.008* | 0.088 |
| Fear | 4.13 (0.98) | 3.19 (0.74) | 4.58 (0.50) | 3.25 (0.66) | 5.32 | 0.29 | 0.024* | 0.065 |
| Anger | 3.90 (0.97) | 2.17 (0.67) | 4.03 (1.07) | 2.58 (0.61) | 1.37 | 0.57 | 0.25 | 0.017 |
| Disgust | 4.00 (1.03) | 2.30 (0.66) | 4.40 (1.01) | 2.20 (0.78) | 2.91 | 0.86 | 0.092 | 0.036 |
| *Totals* | *4.16 (0.95)* | *2.49 (0.70)* | *4.45 (0.75)* | *2.63 (0.63)* | – | – | – | – |

**Notes.**

* Degrees of freedom were always (1, 77).

the summed average of atypical ratings did not overlap for any emotion category. That typicality predictions were supported weakens the suggestion that participants used the internet to generate their responses during study 1. In addition, typicality effects were present for the specific pairings of 'happy'/'smile' and 'sad'/'cry' pairings, reducing the likelihood that participants generated these associations simply as a result of their inclusion in previous task instructions.

## Further manipulations and typicality ratings

### *Rating direction*

The typicality × rating direction × emotion category interaction was significant. Separate typicality × rating direction analyses were conducted for each emotion category (see Table 4).

There were significant typicality × direction rating interactions for the 'sad' and 'fear' categories.

Interactions followed a similar pattern for both emotion categories. As predicted, paired samples $t$-tests showed that participants gave significantly higher likelihood ratings to typical vs. atypical pairs, for both action word-to-category pairings ($t_{sad}(39) = 24.12$, $p < 0.001$, $d = 5.33$; $t_{fear}(39) = 12.74$, $p < 0.001$, $d = 2.30$), and category-to-action word pairings ($t_{sad}(38) = 13.34$, $p < 0.001$, $d = 2.51$; $t_{fear}(38) = 6.98$, $p < 0.001$, $d = 1.10$). Independent samples $t$-tests showed that participants rated atypical pairs similarly, independent of rating direction, ($t_{sad}(74) = -0.079$, $p = 0.94$; $t_{fear}(84) = -0.16$, $p = 0.88$), but gave significantly higher ratings to typical pairings presented in an action word-to-category format than a category-to-action word format, ($t_{sad}(84) = -2.06$, $p = 0.043$, $d = 0.68$; $t_{fear}(84) = -2.004$, $p = 0.048$, $d = 0.59$). In sum, for 'fear' and 'sad' categories, typical pairings were given comparatively higher likelihood ratings when rated in an action word-to-category vs. category-to-action word direction.

**Table 5** **Mean first and third-person perspective ratings, by rating direction, typicality and emotion category.** Standard deviations are presented in parenthesis.

| Emotion category | First person ratings (SD) | | | | Third person ratings (SD) | | | |
|---|---|---|---|---|---|---|---|---|
| | Category-to-Action Word | | Action Word-to-Category | | Category-to-Action Word | | Action Word-to-Category | |
| | Typical | Atypical | Typical | Atypical | Typical | Atypical | Typical | Atypical |
| Happy | 4.70 (0.66) | 1.98 (0.84) | 4.73 (0.45) | 2.22 (0.70) | 4.85 (0.37) | 2.20 (0.68) | 4.85 (0.37) | 2.70 (0.67) |
| Surprise | 3.95 (1.23) | 2.74 (0.95) | 4.00 (1.17) | 2.98 (0.61) | 4.00 (1.12) | 3.19 (0.51) | 4.25 (0.97) | 3.09 (0.42) |
| Sad | 4.05 (1.31) | 1.96 (0.60) | 4.58 (1.03) | 2.15 (0.56) | 4.40 (0.52) | 2.36 (0.54) | 4.70 (0.47) | 2.19 (0.50) |
| Fear | 3.80 (1.20) | 3.02 (0.89) | 4.27 (0.72) | 2.98 (0.79) | 4.45 (0.51) | 3.35 (0.48) | 4.75 (0.44) | 3.50 (0.38) |
| Anger | 3.97 (1.18) | 2.17 (0.80) | 3.92 (1.09) | 2.60 (0.75) | 4.00 (0.73) | 2.16 (0.53) | 4.15 (0.93) | 2.53 (0.47) |
| Disgust | 3.58 (1.22) | 2.39 (0.76) | 4.42 (0.99) | 1.90 (0.75) | 4.40 (0.60) | 2.21 (0.55) | 4.35 (1.23) | 2.45 (0.73) |
| *Totals* | *4.01 (1.13)* | *2.38 (0.81)* | *4.32 (0.91)* | *2.47 (0.69)* | *4.35 (0.61)* | *2.58 (0.55)* | *4.51 (0.74)* | *2.74 (0.53)* |

## Instruction perspective

Critically, there were no significant instruction perspective × typicality, or instruction perspective × typicality × category interactions ($p > 0.10$).

However, the instruction perspective × rating direction × typicality × category interaction was significant (see descriptive statistics in Table 5).

To explore this interaction, separate instruction perspective × typicality × direction rating mixed factorial ANOVAs were conducted for each emotion category. There was a significant interaction for one category only: 'disgust'; $F(1, 82) = 8.71$, MSE $= 0.79$, $p = 0.004$, $\eta_p^2 = 0.097$.[5]

## Discussion

Ratings confirm that participants were more likely to associate action words with the emotional state to which they had been typically generated in study 1. This lessens the likelihood that endorsement was inflated by the examples included in task instructions, or use of the internet. In addition, direction and person perspective manipulations had little impact on ratings, indicating that typical pairings contained action words and emotional states that were robustly associated. Overall, these findings validate the associations derived during study 1 and support the notion that action word meaning is proportionally grounded in, and evokes affective knowledge relevant for different emotional states.

However, it is important to acknowledge the following issue: task design meant that participants rated one label (or action word) in association with all six action words (or labels) before they were presented with the next label (or action word). This may have encouraged participants to adopt a relative rating strategy, in which they simultaneously compared the likely association between all six items and the dominant label, or action word. Typical pairings may then receive the highest likelihood ratings because they represent the 'best option,' rather than giving a true indication of the way in which action words proportionally activate affective knowledge relevant to the presented label. This limitation is compounded as, per block, participants responded to the six pairings in the

[5] To explore this interaction, separate direction rating × typicality mixed ANOVAs were conducted for disgust ratings, for participants who received first and third person instructions, respectively. This two-way interaction was significant for participants who received first-person instructions, $F(1, 37) = 13.06$, MSE $= 0.65$, $p = 0.001$, $\eta_p^2 = 0.26$, but not for those who received third person instructions, $F(1, 37) = 0.45$, MSE $= 0.93$, $p = 0.51$, $\eta_p^2 = 0.012$. Paired samples $t$-tests revealed that, independent of direction rating, participants who had received first person instructions always gave higher ratings to the typical pairing, than grouped atypical pairings, ($t_{\text{category-to-action word}}(18) = 3.90$, $p = 0.001$, $d = 1.20$; $t_{\text{action word-to-category}}(19) = 12.13$, $p < 0.001$, $d = 3.37$.) While independent $t$-tests showed that these participants rated atypical pairings similarly in both rating directions ($t(37) = 1.84$, $p = 0.074$), they gave significantly higher ratings to the typical pairing when embedded in action word-to-category versus category-to-action word sentences, $t(37) = 2.70$, $p = 0.010$, $d = 0.89$.

same order. Any biases that this presentation strategy encouraged would therefore be applicable to all participants, despite care to vary presentation of the typical pair, per block.

However, our data suggest it is unlikely that participants automatically employed a comparative rating strategy. If they had we would expect all atypical pairings to receive very low ratings on the scale. Although some of the averaged, atypical ratings were below the scale midpoint (2.5; 'happy,' 'anger,' 'sad' and 'disgust'), others were higher ('fear' and 'surprise'). These findings are expected given that there were overlaps in the some of the typical action words included in the task and the top, three modal action words generated for other emotion labels, during study 1. This was the case for the three labels that attracted the highest average atypical ratings ('fear,' 'surprise' and 'anger'). For example, although the action word 'cry' represented the typical exemplar for the label 'sad,' it was also frequently generated in response to the emotion labels 'fear' and 'anger' (see Table 1). Similarly, the typical action word for the label 'anger' ('scream') had been frequently endorsed in response to the label 'surprise.' The inclusion of these overlapping exemplars meant that, for some emotion labels, not all 'atypical' exemplars were equally 'atypical,' inflating the averaged atypical ratings. Importantly, these findings indicate that participants judged each action word/label pair based on the 'absolute' association between the two words, rather than making a comparative judgment that was biased by the presence of an obviously 'typical' pairing. They also support the idea that 'typicality' is expressed as a matter of degree, as action words may simultaneously evoke affective knowledge relevant to several emotional states (e.g., *Newcombe et al., 2012*).

One further finding should be highlighted. When direction was manipulated, ratings revealed different forward and backward connection strengths between the emotion labels 'fear' and 'sad' and their paired, typical action words. In both cases participants gave higher ratings when presented with the pair in action word-to-category order, than in category-to-action word order (e.g., P(Hide|Fear) < P(Fear|Hide)). This trend was also present for the label 'disgust' and typical action word of 'recoil,' but only when the pairing was considered from a first-person perspective (see footnote 4).

To aid interpretation we explicitly consider the behaviours to which action words refer, and how they may inform emotional attribution. The present data suggest that the propensity to 'hide' ('cry') when expressing 'fear' ('sadness') may vary depending on the type of stimulus causing 'fear' ('sadness'), but that given the behaviour of hiding (crying), the likelihood that a person is experiencing fear (sadness) is much greater. Arguably the latter attributional pattern may be more prevalent in Western societies. Here people are often encouraged to mask or regulate behavioural signs of emotional states that cause them to be perceived as weak in public, like 'sadness' and 'fear' (e.g., *Wierzbicka, 1994*; *Barrett, Mesquita & Gendron, 2011*). If related behaviours are observed then the attribution process may be more automatic. A justification may follow: 'I/they must be feeling very 'sad' if I/they feel the need to 'cry' in public.' In sum, while the present data confirm that there is stability in the way some action words disproportionately evoke affective knowledge, there is some evidence that cultural background may influence the way affective knowledge is constructed and used for attribution (e.g., *Barrett, Mesquita & Gendron, 2011*).

## GENERAL DISCUSSION

We provide a set of emotion-related action words, accompanied by data to show how strongly each word evokes emotional knowledge relevant to several, discrete emotional states. This work is consistent with the proposal that emotion words are grounded in affective knowledge (e.g., *Vigliocco et al., 2009*) and complements previous research, by exploring whether word-to-knowledge links are constructed, at least partially, in a categorical fashion (e.g.,*Stevenson, Mikels & James, 2007a*; *Briesemeister, Kuchinke & Jacobs, 2011a*; *Briesemeister, Kuchinke & Jacobs, 2011b*; *Briesemeister, Kuchinke & Jacobs, 2014*; *Westbury et al., 2014*).

Action words were elicited from participants using a constrained feature-generation task (e.g., *McRae et al., 2005*; *Vinson & Vigliocco, 2008*). Emotion labels were used to create (and constrain) six, different 'affective contexts' (e.g., *Stevenson, Mikels & James, 2007a*; *Briesemeister, Kuchinke & Jacobs, 2011a*). This method allowed assessment of the strength with which each action word elicited specific affective knowledge; the larger the number of participants who endorsed the pair the greater the likelihood that the word (dis)proportionally evoked knowledge relevant to that emotional state. Using a rating task (study 2) we confirmed that the action words most frequently elicited in study 1 were more likely to be associated with the emotion label to which they had been generated (typical pairs), than to other emotion labels (atypical pairs). Typical pairs also retained rating dominance when two further sentence-based manipulations were applied (rating direction and person perspective), suggesting a degree of robustness in the way typical words evoke affective knowledge.

To facilitate use of the current stimuli, all acceptable action words, generated by two or more participants in study 1, are included in Appendix S1 (a fuller list, including idiosyncratic responses, is provided in the Supplemental Information). Words are presented alongside raw, unmerged frequencies to indicate the number of participants who generated the action word in response to each emotion label. This will allow researchers to select stimuli, based on unmerged frequencies, or apply their own merging criteria. However, for completeness, we also indicate whether the action word was classed as a 'core,' 'subsidiary' (i.e., a synonym for the selected 'core' exemplar) or modal exemplar (a unique, non-synonymous response), based on our merging criteria. Further, we provide ratings for each of the 36 action word/label pairs, included in study 2 (Appendix S2).

On the one hand the current approach, and data produced, may provide an alternative way to select emotional stimuli, based on the extent to which each word is likely to evoke specific affective knowledge (e.g., *Stevenson, Mikels & James, 2007a*; *Briesemeister, Kuchinke & Jacobs, 2014*). The current set of action words may be highly compatible for particular types of task. Previous research shows that participants mimic congruent facial expressions when they encounter emotion words (e.g., *Foroni & Semin, 2009*), and that mimicry leads to enhanced processing of subsequently presented emotional stimuli e.g., valence-congruent sentences (e.g., *Havas, Glenberg & Rinck, 2007*)  and facial expressions (e.g., *Halberstadt et al., 2009*). Based on *Vigliocco et al.*'s (*2009*) framework, we might expect emotion-related action words to more strongly elicit congruent facial

mimicry, given their dual grounding in affective (*Vinson, Ponari & Vigliocco, 2014*) and sensorimotor knowledge (e.g., *Hauk, Johnsrude & Pulvermüller, 2004*; *Niedenthal et al., 2009*; *Moseley et al., 2012*). However, few studies incorporate action words and those that do find inconsistent evidence for a verb (vs. adjective) advantage (*Foroni & Semin, 2009*; *Halberstadt et al., 2009*). If these findings reflect inconsistent use of linguistic stimuli then our data may help by providing a larger set to select from. Further, by choosing words that are both disproportionally related to a particular emotional state and related to facial actions, researchers may extend investigations into whether language-mediated facial mimicry is 'category' or 'valence' driven. Specifically, whether reading an action word strongly associated with 'fear' specifically induces mimicry in category-diagnostic features of a fearful face, (*Ponari et al., 2012*) or whether reading any negatively valenced word induces a similar pattern of negative mimicry.

On the other hand, the present data may encourage two types of additive approach, important for assessing the relative validity of current attempts to measure affective grounding (e.g., *Newcombe et al., 2012*). First, as we provide new categorical data for words which already have dimensional rating norms (e.g., *Warriner, Kuperman & Brysbaert, 2013*), we facilitate attempts to assess whether categorical and dimensional ratings are mutually predictive of one another, or quantify emotional information in the same way. *Stevenson, Mikels & James (2007a)* and *Westbury et al. (2014)* have conducted similar work, both showing a degree of heterogeneity in the ability of categorical ratings to predict dimensional ratings. In particular, *Westbury et al. (2014)* showed that co-occurrence distances between emotion labels and words were more strongly predictive of valence, than arousal ratings, and that both types of dimensional rating were predicted by co-occurrence distances from distinct sets of emotion labels (e.g., those naming 'automatic' emotions, like 'panic,' for arousal, and those associated with approachability and potency, for valence).

This approach could also be used to assess the relationships between the current categorical data and semantic richness norms (e.g., emotional experience ratings), which assess the magnitude to which words evoke undifferentiated, affective knowledge (e.g., *Newcombe et al., 2012*). This is not yet possible, as *Newcombe et al. (2012)* only provide normative data for nouns. It would be particularly interesting to provide a comparison for overlapping exemplars, such as 'cry,' which our participants modally endorsed as evoking affective knowledge relevant to five of the six discrete emotional states. We might expect emotional experience ratings to fluctuate dependent on both the number of emotional states that the word can be associated with, and the frequency of endorsement, across emotions.

A second, related investigation, would involve entering different types of rating as separate predictors, to assess whether they account for unique variance in emotional word processing outcomes. Previous work focuses on prediction of lexical decision latencies, presumably because large datasets of reaction times already exist (e.g., *Balota et al., 2007*; *Keuleers et al., 2012*). However, it may be equally possible to apply ratings as predictors to other types of task that examine emotional word processing (*Briesemeister, Kuchinke & Jacobs, 2011b*). For example, the emotional Stroop task (*MacKay et al., 2004*) and

*De Houwer*'s (*2003*) affective Simon task (*Altarriba & Basnight-Brown, 2010*). So far, lexical decision data confirm that categorical and dimensional ratings account for unique variance and that, when combined, ratings account for a slightly larger proportion of overall variance in latencies than they do independently (e.g., *Briesemeister, Kuchinke & Jacobs, 2011a*; *Briesemeister, Kuchinke & Jacobs, 2011b*; *Briesemeister, Kuchinke & Jacobs, 2014*; see also *Newcombe et al., 2012* and *Moffat et al., 2014* for comparisons of semantic richness and dimensional ratings). In support, physiological evidence shows that both types of information are important for emotion word processing; when words are disproportionately associated with particular emotional states then categorical information is processed first, followed by dimensional or valence-based properties of the word (e.g., *Briesemeister, Kuchinke & Jacobs, 2014*; *Briesemeister et al., 2014*). Linear processing stages are consistent with *Panksepp*'s (*1998*; *2012*) hierarchical model, which includes a secondary, automatic stage for categorical processing of emotional stimuli (related to the proposed play, seeking, rage, lust, fear, panic and care subsystems), and a subsequent, tertiary stage, in which dimensional properties of the stimuli are considered.

However, one caveat is important when considering the compatibility of our stimuli for lexical decision, or other tasks that require single-word processing. As previously argued, participants tend to be poor or inconsistent in their ability to extract affective meaning from verbs (e.g., *Schacht & Sommer, 2009*; *Palazova et al., 2011*). Meaning activation will depend on the task in which the verb is presented, and its associated goals. For example, when action words are presented in isolation and participants make a non-affective judgment, as they do in lexical decision tasks, action words are unlikely to spontaneously evoke the same constrained, affective knowledge that they do in the present work. Therefore, in order to assess whether categorical ratings predict action word processing, the same 'affective context' might need to be applied to the new task. Following *Schacht & Sommer*'s (*2009*) approach, researchers might present the word pair 'sad' and 'cry,' asking participants to respond to the action word in the pair, only.

Situated approaches emphasise that words are referents for experiential components; in this case, behaviours. As such, some of our findings have implications for how overt cues influence emotional attribution and interpretation. Finding that participants sometimes associated the same behaviours with several emotional states in study 1, and showed fluctuations in their ratings of atypical behaviour/state pairings in study 2, both stand in contrast to basic emotion views (e.g., *Ekman, 1992*). These accounts suggest that behaviours show strong, discrete, relationships with basic emotional states and are important diagnostic cues for interpretation. In contrast, proportional associations are favoured both by construction and componential models (e.g., *Scherer, 1984*; *Smith & Ellsworth, 1985*; *Barrett, Lindquist & Gendron, 2007*; *Lindquist, 2009*). According to these accounts, behavioural cues need not be diagnostic as emotional interpretation is driven by the summation of multiple pieces of evidence, only some of which will be present at the time of perception (e.g., *Smith & Ellsworth, 1985*; *Lindquist & Gendron, 2013*). People flexibly recruit other 'evidence' from a highly intra-individual repository of affective knowledge, built through relevant past and present experiences (e.g., what precipitated

the current emotional state, how the actor has behaved in the past, how the observer themselves felt under similar circumstances). Some of this knowledge will be shaped by the societal or cultural norms applicable to the individual (see study 2, e.g., *Barrett, Mesquita & Gendron, 2011*). Flexible knowledge recruitment explains why the same behaviour may be interpreted to represent different emotional states by different observers, or by the same observer, across different time-points (e.g., *Lindquist & Gendron, 2013*).

In conclusion, we provide a set of English action words, characterised by their proportional likelihood to evoke affective knowledge relevant to different emotional states. We used basic emotion labels to create a set of constrained 'affective contexts,' both for initial generation of action words (study 1) and validation of the most typical exemplars (study 2). Our stimuli both complement and extend existing linguistic databases that contain categorical norms (e.g., *Stevenson, Mikels & James, 2007a*; *Briesemeister, Kuchinke & Jacobs, 2011a*). Our data may similarly be used to explore whether emotional word processing is predicted by categorical norms alone, or in conjunction with other types of rating (e.g., dimensional or semantic richness ratings, *Lang, 1980*; *Newcombe et al., 2012*).

### Funding

This work was supported by an ESRC studentship, awarded to the Emma Portch. The funders had no role in study design, data collection and analysis, decision to publish, or preparation of the manuscript.

### Grant Disclosures

The following grant information was disclosed by the authors:
ESRC.

### Competing Interests

The authors declare there are no competing interests.

### Author Contributions

- Emma Portch conceived and designed the experiments, performed the experiments, analyzed the data, wrote the paper, prepared figures and/or tables, reviewed drafts of the paper.
- Jelena Havelka conceived and designed the experiments, wrote the paper, reviewed drafts of the paper.
- Charity Brown and Roger Giner-Sorolla conceived and designed the experiments, reviewed drafts of the paper.

### Human Ethics

The following information was supplied relating to ethical approvals (i.e., approving body and any reference numbers):

Approval was obtained from the School of Psychology's internal ethics committee at the University of Leeds. This committee follows the guidelines set out by the British

Psychological Society's code of conduct, ethical principles and guidelines (1993). The research was approved on the 24th of February, 2013 and assigned the following ethics reference: 13-0032. Both study 1 and 2 were approved under the same ethics code (application title: 'Processing different emotion words: obtaining affective norms for British English'). Approval was acknowledged via email.

## Supplemental Information

Supplemental information for this article can be found online at http://dx.doi.org/10.7717/peerj.1100#supplemental-information.

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
