# Peer review of "Using affective knowledge to generate and validate a set of emotion-related, action words"

_PeerJ, doi:10.7717/peerj.1100_

## Round 0.1 · original submission · Minor Revisions

Dear Dr. Portch,

I have now received 4 reviews of your paper. I thank the reviewers for their work. Even if there are differences between the reviewers' final assessment - their evaluations vary from "accept" (Reviewer 1) to "major revision" (Reviewer 4)-, I think that all reviewers converge in recognizing that your work is interesting and methodologically sound. Based on my own independent reading of your manuscript, I agree with them.

For a successful revision, I think you will need to better ground your work in the recent literature, referring to current theories on lexical representation and emotion (see comments of Reviewer 4) and to further recent work on a similar topic (Reviewers 2 and 3). You would also need to recognize possible limitations of your study: for the fact that the role of contexts is not so strong (see Reviewer 4) and the fact that absence of randomization can bias your results (Reviewer 2).
Thank you for submitting your interesting work to PeerJ. I am looking forward to your revised paper.

Sincerely,
Anna M. Borghi

Reviewer 1 ·

Basic reporting

The ms presents two empirical studies in which action verbs were generated and rated for each of Ekman basic emotions. The norms are useful to the research community. The design and data analysis appear competently handled.

Experimental design

Adequate

Validity of the findings

Research is descriptive

Reviewer 2 ·

Basic reporting

The article is well written, nicely introduced and easy to follow. I only have one question regarding the the fitting into the broader field.
A recent study by Westbury, Keith, Briesemeister, Hofmann & Jacobs (2014) documents that it is possible to predict emotion ratings for words on the basis of co-occurance statistics and their distance to other emotion words. I think this nicely relates to the idea of context dependent scripts for emotion words and I think it would be interesting to see how this idea of an algorithmic qunatification of context relates to the present manuscript.


Reference:
Westbury, C., Keith, J., Briesemeister, B. B., Hofmann, M. J. & Jacobs, A. M. (2014). Avoid violence, rioting, and outrage; approach celebration, delight, and strength: Using large text corpora to compute valence, arousal, and the basic emotions. The Quarterly Journal of Experimental Psychology, DOI: 10.1080/17470218.2014.970204

Experimental design

In general, the experimental idea and design are well done. I only have one minor question regarding the experimental design of study 2, where the item presentation was not randomized. I wonder whether there is a reason for this, given that priming effects might bias the results when stimuli are presented in a non-randomized way.
When there is no design-theoretic reason, I would like to see at least a short discussion on the possibility of biased results because of priming and/or sequential effects.

Validity of the findings

no comments

Reviewer 3 ·

Basic reporting

Overall this a well written manuscript, presenting two studies that examine emotional scripts and their relation to discrete emotion categories. The authors further examined the robustness of their obtained association strenghts between (most frequently generated) verbs and the discrete emotion categories. Most importantly the full verb generation data are provided in the appendix to stimulate future reseach.
I would only like to ask the authors to also include a reference to the recent attempt by Westbury et al (2014) to generate quite similar data based on large text corpora, and discuss how this approach differs from the recent study.

*Westbury, C., Keith, J., Briesemeister, B. B., Hofmann, M. J. & Jacobs, A. M. (2014). Avoid violence, rioting and outrage; approach celebration, delight, and strength: Using large text corpora to compute valence, arousal, and the basic emotions. Quarterly Journal of Experimental Psychology.

Experimental design

The manuscript describes original research and the methods used are sound. Still, it could be made clearer why synonyms were omitted (or merged) from the analyses?

Also, i would like the authors to further validate their examination of the typicality effects, as in the present analyses, the data of 1 examplar is compare with the averaged data of five other associations (in study 2). This could likely have biased the results and it would be interesting to read whether a typicality effect is still visible when the typical and the highest atypical rating are compared? Or some data is provided that shows that the confidence interval of the mean rating of the atypical does not overlap with the ratings of the typical verbs...

Validity of the findings

see above

Reviewer 4 ·

Basic reporting

The submission is self-contained, generally coherent and conforms to professional standards (although there are many typos that need to be corrected in revision). Some essential details need to be expanded on, or clarified in the introduction/discussion:

A general comment: It would be especially helpful to explain more clearly how script theory relates to theories of lexical representation and processing. Especially as there are (suddenly) a number of views of lexical processing that take emotional content into account, mostly variants of the idea that words with emotional connotations evoke those emotions in the comprehender thus influencing lexical processing to some extent (some prominent authors taking these directions include Pexman, Pulvermuller, Vigliocco, Yap, and even the most recent incarnation of Paivio's dual-coding theory). In particular, script theory seems to especially resonate with the idea that words' meanings depend deeply on various aspects of physical experience including perception, action and emotion (often couched in terms of conceptual features, such as that apples are red, hammers can be manipulated by holding with a single hand, and crying is evoked by sadness). Does script theory provide a more precise mechanism by which these links between language and emotion may affect processing of single words?

In the introduction the authors especially argue for the importance of context in the comprehension/interpretation of emotional words, especially verbs. In particular, it is difficult/inappropriate to interpret typical ratings of words in terms of a decontextualised dimensional model (valence-arousal) and that context needs to be taken into account. However I am not entirely convinced that collecting ratings of words' association with basic emotion terms is truly suitable for overcoming this concern. As the authors discuss elsewhere in the introduction (eg. lines 72+) effects for single words differ greatly depending on sentence context, undermining the present approach in which associations between single words (e.g. SADNESS - CRY) are taken as "contextual". And the sentences that are provided in study 2 (see line 285) provide very little context beyond the pairing of the two words in question: they are nearly semantically empty and constant across the items tested. While this is necessary for experimental control, it weakens the argument that the present approach is suitably contextual.

There is also some evidence that appears to support the dimensional model of emotion (e.g. Kuperman et al, Journal of Experimental Psychology: General 2014; Kuperman, Quarterly Journal of Exp Psychology 2014) so it may be premature to reject this approach. In my view such findings show that despite strong effects of sentence context (and other kinds of contexts), there remain some strong, reliable patterns of emotion in lexical processing even when single words are rated in a decontextualised manner - or presented out of context in a single-word experiment.

Minor points:
Lines 43-47: the authors mention that 44.54% of words were strongly related to one or more discrete emotion labels in Stevenson et al (2007). Can it be clarified what kind of criterion was used here? It's not obvious from Stevenson et al what this was - and how it relates to the idea that discrete emotion categories are useful for providing specific contexts?

Verbs vs action words: it's also important to take note that not all action words are verbs: action nouns exist and tend to have extremely similar semantic makeup to their verb counterparts.

Experimental design

Methods are described appropriately, with sufficient level of detail throughout and in line with ethical standards.

Validity of the findings

The authors used a highly constrained paired-associate generation task (Basic emotion word -> Verb). how does this differ from unconstrained free association as in Nelson et al (2004)? Also relevant in this regard is semantic feature generation: see Buchanan et al (Behavior Research Methods 2013) in which several of the terms used here are employed in an unrestricted semantic feature generation task --- typically corresponding to the patterns observed in the present Study 1 which I think provides further, if indirect, validation of the present approach.

The authors seem to go back and forth in discussing "discrete" emotions vs. their overlap. Sometimes this is discussed as a strong distinction, but other times as a matter of degree. For example, in some places the fact that individual words appear across multiple basic emotions is taken as evidence that they should be considered as homonyms: different words that happen to have the same form (the case of "jump" on lines 194+) but in other places a different approach is taken (the case of "cry" which does not seem to attract similar treatment despite being produced regularly for multiple basic emotions). First of all, I would like more information about the criteria for cases that were merged vs. unmerged (like JUMP), and how frequently this occurred. And secondly, are cases like "cry" problematic for the idea that basic emotions are discrete? This issue is relevant across the manuscript: the introduction, decisions made in data modifications (study 1) and the interpretation of the results, and I think the authors need to be more explicit about it.

One very important decision the authors made, was to combine synonyms in study 1 (starting about lines 186). I am worried about this procedure, especially if one of the aims is to provide normative materials that can be used in future studies. If production of "shout" + "yell" is combined into a single case, it seems to me that the production frequency of "shout" will be an overestimate of the link between a basic emotion like ANGER and the specific word "shout". This is especially problematic for the next example provided: cases like "laugh" + "giggle" treated as synonymous based on thesaurus co-occurrence in both directions. Consulting this particular source (dictionary.cambridge.org) it appears that laughing and smiling are combined into a set of "synonyms, related words and phrases that make up this topic." I would prefer that these decisions are checked and confirmed using a more conventional thesaurus that does not also include related words. These seem to be especially made up of cases where a more specific term (subordinate/troponym) is combined with its superordinate/hypernym, which implies some important assumptions about the categorical structure of verbs.

I also noticed that a very large number of responses were idiosyncratic, nearly 70% of the responses that were deemed acceptable according to the exclusion criteria. This seems like a wealth of data that could be potentially useful - as a vetted set of verbs that meet the criteria for selection. I suggest that this list is provided as additional supplemental materials.

For Study 2, the design is highly appropriate to serve as validation of the results from Study 1: mainly the overall consistently high ratings for the most typical verb-emotion pairing.
I am particularly interested in the apparent variability among the atypical items. If these verbs were truly diagnostic of a particular emotion, or at least extremely associated with it, I might have expected them to be at floor when rated for other emotions. After all, the very top word for each emotion was selected from study 1. This is especially the case for surprise and fear, for which the mean of all five "atypical" verbs is greater than 3, the midpoint of the 1-5 rating scale. Is this problematic for a neatly discrete account of emotions and associated verbs?


Minor points:

Line 128-129 The association task used in study 1 differs importantly from the feature generation methods used by Cree & McRae (2003) and Vigliocco et al (2004) - both explicitly instructed participants to avoid free associations.

Line 172: The study was conducted online and participants were "discouraged from using the internet" to do the task. Is there any way of ruling out this possibility - if they did it anyway, it would very significantly undermine the validity of the study. I think the authors can/should defend themselves against this possibility: diversity of responses across participants suggests this was not a common problem, and the validation provided by study 2 further supports this.

Line 182: "confused" is mentioned as a case that should be excluded as a synonym for the emotion label itself. Which emotion label is "confused" a synonym of? (perhaps the wrong example was mentioned here - I think "confused" fits under exclusion criterion (a) as it's far more likely to be an adjective rather than a past-tense verb).

Lines 252-254: this seems very vague, can the authors make this more specific?

Lines 290-294: it would be useful to discuss the difference between forward and backward associations in terms of conditional probabilities. This is especially important given the differences in directionality that were observed for sadness/fear and which are not very clearly discussed (lines 438-453): for example the propensity to hide when expressing fear may vary depending on the type of stimulus causing fear, but that given behaviour of hiding, likelihood that a person is experiencing fear is much greater: P(Hide|Fear) < P(Fear|Hide).

Line 382: What is meant by "summed atypical pairings"? It looks to me like the averages, not the sums were compared to the typical parings.

Line 466-468: can it really be said that the present task mimics real-world situations in which people use scripts to make emotional attributions? The present task seems excessively constrained for this to be the case (although I agree this is necessary for experimental control).

Various places: the authors often report a high level of precision (e.g. line 215: 21.82% of unique words) - often too many decimal places are reported given the number of observations.

---

## Round 0.2 · accepted · Accept

I am happy to inform you that your manuscript has been accepted for publication in PeerJ.

Reviewer 2 ·

Basic reporting

no comments

Experimental design

no comments

Validity of the findings

no comments

Additional comments

Nice work! I really enjoyed it.

Reviewer 4 ·

Basic reporting

Meets all standards.

Experimental design

Meets all standards, and the level of detail is particularly commendable.

Validity of the findings

Meets all standards.

Additional comments

The authors have made very substantial revisions that fully address the main concerns I raised (and go beyond them in several cases). The amount of detailed information available for users in the form of various angles of analysis and detailed supplemental files is particularly valuable as this allows full critical evaluation of all aspects of the study and corresponding dataset by any user.

Even though I still have personal concerns about the fairly limited scope of the particular words used for the generation task, I see this work as having plenty of potential utility, especially among the numerous researchers who are doing work around these "basic" emotions.